# Thromboelastography Profile Is Associated with Lung Aeration Assessed by Point-of-Care Ultrasound in COVID-19 Critically Ill Patients: An Observational Retrospective Study

**DOI:** 10.3390/healthcare10071168

**Published:** 2022-06-22

**Authors:** Daniele Guerino Biasucci, Maria Grazia Bocci, Danilo Buonsenso, Luca Pisapia, Ludovica Maria Consalvo, Joel Vargas, Domenico Luca Grieco, Gennaro De Pascale, Massimo Antonelli

**Affiliations:** 1Department of Emergency, Anesthesiology and Intensive Care Medicine, Fondazione Policlinico Universitario “A. Gemelli” IRCCS, Largo “A.Gemelli” 8, 00168 Rome, Italy; mariagraziabocci@gmail.com (M.G.B.); luca.pisapia92@gmail.com (L.P.); vargas.joel87@gmail.com (J.V.); dlgrieco@outlook.it (D.L.G.); gennaro.depascale@policlinicogemelli.it (G.D.P.); massimo.antonelli@unicatt.it (M.A.); 2Department of Woman and Child Health and Public Health, Fondazione Policlinico Universitario “A. Gemelli” IRCCS, Largo “A.Gemelli” 8, 00168 Rome, Italy; danilo.buonsenso@policlinicogemelli.it; 3Anesthesiology and Intensive Care, Fondazione IRCCS Ca’ Granda Ospedale Maggiore Policlinico, Via Francesco Sforza 35, 20122 Milan, Italy; marycons24@gmail.com

**Keywords:** lung ultrasound, thromboelastography, hypercoagulability, COVID-19, SARS-CoV-2

## Abstract

**Background.** To evaluate relationships between lung aeration assessed by lung ultrasound (LUS) with viscoelastic profiles obtained by thromboelastography (TEG) in COVID-19 respiratory failure. **Methods.** Retrospective analysis in a tertiary ICU in Rome, Italy. Forty invasively ventilated adults with COVID-19 underwent LUS and TEG assessment. A simplified LUS protocol consisting in scanning six areas, three per side, was adopted. A score from 0 to 3 was assigned to each area. TEG^®^6s was used to obtain viscoelastic hemostatic assay parameters which were compared to LUS score. **Results.** There was a significant inverse correlation between LUS score and static compliance of the respiratory system (Crs, rs −0.75; *p* < 0.001). We found a significant association between LUS and functional fibrinogen maximum amplitude (FF-MA): among 18 patients with LUS score ≤ 12, median FF-MA was 31 mm [IQR 28–39] whilst, among 22 patients with LUS score > 12, it was 46.3 mm [IQR 40–53], *p* = 0.0004. Median of the citrated recalcified kaolin-activated maximum amplitude (CK-MA) was 66.1 mm [64.4–68] in the LUS score ≤ 12 group, and 69.6 [68.5–70.7] when LUS score > 12, *p* < 0.002. **Conclusions.** The hypercoagulable profile as defined by elevated FF-MA and CK-MA may be associated with a low degree of lung aeration as assessed by LUS.

## 1. Introduction

In critically ill patients with confirmed coronavirus 2019 disease (COVID-19), laboratory abnormalities compatible with hypercoagulability and a high prevalence of thromboembolic events have been widely reported [1,2,3,4], with evidence of both macro- and microthrombotic events concerning both venous and arterial districts [5,6,7].

Authors have mainly reported a hypercoagulable profile on ICU admission with increased D-dimer levels up to 10 times the upper limit of the normal range, increased fibrinogen levels and enhanced platelet activation [8,9]. Furthermore, an increased clot strength and a hypercoagulability state have been assessed by viscoelastic methods such as thromboelastography (TEG) [8,10,11,12].

The crosstalk between the immune system and the coagulation system underlies the process of microvascular clot formation and microangiopathy found in the lung and other organs [13]. The mechanisms leading to these severe dysfunctions are a virus direct endothelial injury and a leucocyte- and cytokine-mediated endothelial injury and a procoagulant state characterized by an increased release of tissue factor, platelets and complement activation and inhibition of fibrinolysis and anticoagulation pathways. This framework configures the so called “COVID-19-associated coagulopathy” that is currently known to be pivotal in the pathophysiology of SARS-CoV-2 infection [14,15].

On the other hand, point-of-care lung ultrasound (LUS) has been proposed and used as an alternative to chest radiography for diagnosis and monitoring of COVID-19-related interstitial pneumonia and acute respiratory distress syndrome, with the advantages of easier bedside use, easy repeatability, immediate availability, fewer operators exposed to the virus and to ionizing radiations during the single exam [16,17]. Several studies have proved that in SARS-CoV-2 infection LUS is predictive of pneumonia severity, as assessed by chest computed tomography and clinical features [16,17,18], and its use as a diagnostic and monitoring tool is supported by the newest consensus statements [19]. Furthermore, in COVID-19 hospitalized patients, the LUS score is strongly associated with the need for invasive or non-invasive mechanical ventilation and is a good predictor of mortality [18,19,20,21].

Laboratory findings suggesting a hypercoagulability state, such as D-dimer elevation, have been proved to be associated with disease severity, radiological lung involvement as on computed tomography scan and outcome [22,23]. 

However, to our knowledge, no data have been published yet describing the relationships between LUS findings and procoagulant profiles in COVID-19 adult critically ill patients.

Thus, this study was aimed at determining the relationships between lung aeration assessed by LUS with hypercoagulability profiles and severity as assessed by TEG. 

## 2. Materials and Methods

This is an observational retrospective study, which was conducted, following STROBE guidelines, at the Fondazione Policlinico Universitario “A. Gemelli” IRCCS in Rome, Italy. All procedures performed in this study were in accordance with the ethical standards of the Institutional Ethics Committee which approved the study (Protocol N° 0016763/20, Protocol ID 3146) and with the 1964 Helsinki Declaration and its later amendments or comparable ethical standards.

A cohort of adult patients consecutively admitted to our intensive care unit from 6 March to 30 April 2020, undergoing invasive mechanical ventilation due to confirmed severe COVID-19 infection and screened by both early LUS and TEG before starting any kind of antiviral, antithrombotic or corticosteroid treatment, were included in the analysis. Patients were excluded in the case of previously established diagnosis of heart failure, interstitial lung disease of further etiology, long-term home oxygen therapy or denying informed consent.

The LUS exam was carried out during patients’ initial assessment by five physicians proficient in critical care ultrasound and by one nurse trained in protocol-based image acquisition. A wireless ultrasound device equipped with a convex probe (ATL s.r.l., Milan, Italy) was used. To minimize contagion risks for healthcare providers and to reduce time, we adopted a simplified LUS protocol, modified from those proposed by other authors [24,25], and already validated in COVID-19 patients in a previous article from our research group [20]. The simplified LUS protocol which was adopted consists in scanning a total of 6 lung areas, 3 for each lung: the scanned areas are limited by the inter-mammillary line (IML) and by the parasternal (PSL), mid-clavicular (MCL), anterior axillary (AAL), middle axillary (MAL) and posterior axillary lines (PAL). Right anterior upper and left anterior upper areas (non-dependent areas) were evaluated on MCL above the IML. Right lateral lower and left lateral lower areas were evaluated by placing the probe just below the IML and between AAL and MAL. Finally, right posterior lower and left posterior lower areas (dependent areas) were evaluated just above the diaphragm by placing the probe along PAL. A LUS score from 0 to 3 was assigned to each lung area. Comprehensive LUS score was calculated as the sum of the score in the 6 lung areas. A *0* indicates a normal lung with the presence of lung sliding: visible A-lines with fewer than 3 B-lines per intercostal space (non-involved area). A *1* represents a small loss of aeration characterized by more than 3 B-lines or the presence of multiple sub-pleuric consolidations separated by normal pleura. A *2* indicates a moderate loss of aeration consisting in multiple and coalescent B-lines and/or multiple sub-pleuric consolidations 1 × 2 cm or smaller and separated by thickened or irregular pleura. A *3* depicts a severe loss of aeration described as parenchymal consolidation or sub-pleuric consolidations greater than 1 × 2 cm.

To assess hemostasis and coagulation processes, we have used a TEG^®^6s system (Haemonetics, Braintree, MA, USA), allowing us to perform real-time analysis of viscoelastic properties of the clot-forming. TEG^®^6s enables multiple assays to be performed simultaneously from a single blood sample. We recorded the first sample collected per patient, within 24 h after ICU admission. The following tests were performed: (1) kaolin TEG (CK), which describes the intrinsic pathway activation and can assess the overall coagulation function; (2) kaolin TEG with heparinase (CKH) which neutralizes the effects of heparin and can assess the presence of heparin or heparinoids; (3) rapid TEG (CRT), which describes the extrinsic pathway using both kaolin and tissue factor; (4) TEG functional fibrinogen (CFF), which uses tissue factor as coagulation activator and GpIIb/IIIa inhibitors to neutralize platelet function in order to measure the fibrinogen contribution to clot formation.

Ventilation settings were as follows: volume-controlled mode with tidal volume 6 mL·kg^−1^ of predicted body weight, inspiratory flow 60 L·min^−1^, inspiratory pause 0.3 s, respiratory rate titrated to obtain pH between 7.35 and 7.45 with <35 breaths per minute, FiO2 titrated to a SpO2 between 90% and 96%. The static compliance (Crs), obtained from a standard 0.3 s inspiratory pause, was calculated as tidal volume/(end-inspiratory plateau pressure–total PEEP) 15 min before starting LUS examination.

Demographics and relevant comorbidities were recorded within the first 24 h of ICU admission. Blood samples for standard hematological parameters, including platelet count, were obtained from an arterial line on ICU admission. Two blood citrate samples (3 mL) were also collected for laboratory standard coagulation tests such as activated partial thromboplastin time (aPTT), prothrombin time (PT), international normalized ratio (INR), D-dimer levels and antithrombin. DIC scores per International Society on Thrombosis and Haemostasis (ISTH) criteria and sequential organ failure assessment (SOFA) score were calculated for each patient on ICU admission. Heart rate, level of consciousness, oxygenation and respiratory rate were recorded on admission. The ratio of SpO_2_ to the fraction of inspired oxygen (SpO_2_/Fio_2_), arterial blood gases (PaO_2_; PaCO_2_), pH and PaO_2_/FiO_2_ ratio were also recorded. A chest X-ray was also taken on admission and its findings were also collected.

### Statistical Analysis

Normal distribution of continuous variables was assessed by Shapiro–Wilk test. Continuous variables with normal distribution are presented as mean ± standard deviation (SD) and were compared using Student’s *t*-test. Continuous variables with non-normal distribution are presented as median and interquartile range (IQR) with 95% CIs and were compared by the non-parametric Wilcoxon rank-sum test. Categorical variables were presented as number of patients (percentages) and were compared using the χ2 or Fisher’s exact test, as appropriate. No statistical sample size calculation was performed a priori, and the sample size was equal to the number of patients treated during the study period.

LUS score was compared to the static compliance of the respiratory system (Crs) at ICU admission using Spearman correlation coefficients (rs). LUS score > 12 was considered as associated with worse severity of the pulmonary disease, since it has been proved to be the most accurate cut-off value for worse outcome prediction in a previously published study [20]. Inter-group comparisons between low and high LUS score (>12) and low and high Crs were performed with the non-parametric Wilcoxon rank-sum test.

A *p* ≤ 0.05 was considered statistically significant. Statistical analysis was performed using Stata/BE 17 (StataCorp LLC, 5 Lakeway Dr, College Station, TX, USA). The analyses have not been adjusted for multiple comparisons, and given the possibility of a type I error, the findings should be interpreted as exploratory and descriptive.

## 3. Results

A total of 40 patients were included in the analysis whose demographics and laboratory characteristics are detailed in Table 1.

Median static compliance of the respiratory system was 37 mL/cm H2O (53.6 mL/cm H_2_O [IQR 42–65.7] in 18 patients with LUS ≤ 12; 32.5 mL/cm H_2_O [23.3–37] in 22 patients with LUS > 12, *p* < 0.001, Table 1.

There was a significant and strong correlation between LUS score and Crs: the lower the Crs the higher the LUS score, rs −0.75; *p* < 0.001 (Figure 1A). In fact, LUS score > 12 was strongly associated with Crs ≤ 37, *p*< 0.001 (Figure 1B).

We found a significant and strong association between LUS score and functional fibrinogen maximum amplitude (FF-MA) which is representative of the maximum amount of clot strength from fibrin (Figure 2A). Among 18 patients with LUS score ≤ 12, median FF-MA (range of normal values: 15–32 mm) was 31 mm [IQR 28–39], whilst among 22 patients with LUS score > 12, median FF-MA was 46.3 mm [IQR 40–53], *p* = 0.0004 (Table 1 and Figure 2A). There was also a significant but moderate association between Crs and FF-MA, *p* = 0.03 (Figure 2B).

The median of the citrated recalcified kaolin-activated maximum amplitude (CK-MA, range of normal values: 52–69 mm), representative of the maximum clot strength due to platelet and fibrin interaction, was 66.1 mm [64.4–68] among patients with LUS score ≤ 12, and 69.6 [68.5–70.7] among those with LUS score > 12, *p* < 0.002 (Table 1).

## 4. Discussion

According to previous findings [10,26], data from the present study confirm a procoagulant condition detectable with both common laboratory tests and viscoelastic methods, mainly related to fibrinogen levels and function, as evidenced by elevated FF-MA values (Table 1 and Figure 2), elevated CK-MA and increased maximum amount of citrated functional fibrinogen at 10 min after clotting time (CFF-A10) (Table 1).

In fact, the severe inflammatory state seen in critically ill adult patients with COVID-19 and the related “COVID-19-associated coagulopathy” can determine endothelial injury and a subsequent procoagulant condition, increasing the risk of perfusion deficit in micro- and macrocirculation and contributing to hypoxemia and to disease severity [27,28].

While it has been established that elevation of fibrin degradation products (D-dimer) is correlated with higher probability of pulmonary embolism, with disease severity and worst outcome [29], only a few data are available on the relationship between viscoelastic methods and prognosis. 

What is new from our study is that we found a positive and significant association between LUS score and a procoagulant profile as strongly correlated with disease severity. Furthermore, in our small cohort of COVID-19 patients, we found that LUS score is inversely correlated with lung compliance, confirming previous findings on different lung pathologies and non-COVID-19-related hypoxemic respiratory failure [30,31]. In other words, since LUS has been positively correlated with chest computed tomography findings and inversely correlated with lung aeration [30,31], the higher the LUS, the lower the lung aeration, and the worse the procoagulant state which may account for thromboembolic events and disease worsening. 

Unfortunately, during the massive surge of COVID-19 critical patients, we were not able to perform chest CT for all patients. Among 22 patients with LUS score > 12, only four patients underwent a contrast-enhanced CT scan documenting a condition of diffuse micro- or macropulmonary embolism which we were able to hypothesize using a multimodal assessment based on the integration between LUS and TEG.

Previous findings by Grasselli et al. showed how patients with a low pulmonary compliance and high levels of D-dimer, underlying a procoagulant state, have a higher risk of mortality [32]. Few data are available on using LUS for lung compliance assessment [31], however, based on our findings and available evidence, we can suppose that lung ultrasound can be a good tool for evaluating lung aeration and its integration with a patient’s coagulation state can be useful for assessing severity and outcome.

Our preliminary data suggest that in COVID-19-related respiratory failure, a high LUS, which reflects a poor aeration, is associated with the highest FF-MA values on TEG, which reflects a hypercoagulability state. Thus, a combined assessment of lung aeration by LUS and hemostatic and viscoelastic profile by TEG may constitute a powerful point-of-care tool for assessing severity and prognosis and for bedside monitoring of COVID-19 critically ill patients, since contrast-enhanced chest CT may be a limited option during a massive surge of COVID-19 critical patients. 

This study has several limitations represented by its observational retrospective nature at a single center. These observations on a small cohort of 40 patients need confirmation by larger trials aimed at comparing LUS score and FF-MA values with lung tissue density and thromboembolic events determined through chest computed tomography. The findings of this pilot study should be interpreted with caution, considering its purely exploratory and descriptive nature.

## 5. Conclusions

Our data suggest that in COVID-19 respiratory failure, a loss of lung aeration assessed using LUS may be associated with a hypercoagulability state as defined by high FF-MA values. 

During pandemics, and particularly during those periods of massive surges of patients to emergency wards, it is crucial to use point-of-care tools for rapid assessments, avoiding resorting to traditional diagnostic instruments (i.e., CT scan, standard laboratory tests), which may take more time to give results and have limited availability and difficult logistics. 

Integration of TEG assay and LUS, easily available at bedside, can be useful and powerful tools for severity assessment and monitoring of high-risk COVID-19 patients. Further prospective studies are needed to eventually confirm these preliminary findings.

## Figures and Tables

**Figure 1 healthcare-10-01168-f001:**
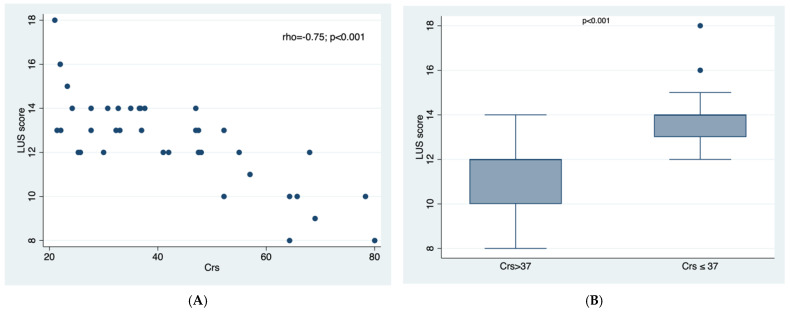
Figures representing the relationships between LUS score and static compliance (Crs). (**A**). LUS score and Crs are inversely correlated, the higher the LUS, the lower the Crs. (**B**). Low compliance group (Crs ≤ 37) has been found to be associated with the highest LUS scores.

**Figure 2 healthcare-10-01168-f002:**
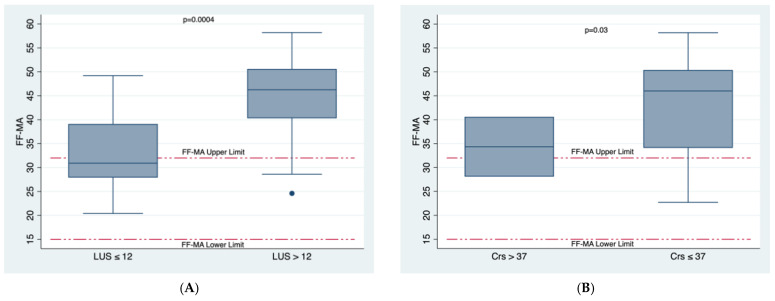
Figures showing FF-MA relationships with LUS score and Crs. (**A**). Significant and strong association between LUS score and FF-MA which is representative of the maximum amount of clot strength from fibrin, *p* = 0.0004. (**B**). There was also a significant but moderate association between Crs and FF-MA, *p* = 0.03.

**Table 1 healthcare-10-01168-t001:** **Demographic, Clinical and Laboratory Characteristics.** Data are expressed as: median [interquartile range, IQR], frequencies (%). DIC and SOFA scores are expressed as mean ± standard deviation (±SD).

	LUS Score ≤ 12 (N = 18)	LUS Score > 12 (N = 22)	*p*
**Sex, female. No. (%)**	5 (27.8%)	7 (31.8%)	0.92
**Age [IQR]**	60 [48–77]	75 [56–79]	** *0.09* **
**Cardiovascular disease. No. (%)**	9 (50)	12 (63)	0.4
**Respiratory disease. No. (%)**	2 (11)	5 (26)	0.2
**Diabetes. No. (%)**	3 (16)	2 (10)	0.5
**SpO_2_/FiO_2_. Ratio [IQR]**	204 [95–423]	188 [126–196]	0.84
**Crs [IQR]**	53.6 [42–65.7]	32.5 [23.3–37]	** *<0.001* **
**DIC score. Mean (±SD)**	2.7 (0.7)	3.2 (0.4)	** *<0.01* **
**SOFA score. Mean (±SD)**	4 (2.7)	7 (2.4)	** *<0.01* **
**aPTT. Sec [IQR]**	33.7 [28.7–44.8]	35.4 [31.3–41.9]	0.46
**INR. Ratio [IQR]**	1.07 [1–1.15]	1.11 [1.03–1.21]	0.29
**Fib. mg/dL [IQR]**	487 [385–596]	557 [428–756]	0.37
**Platelet count. × 10^9^/L [IQR]**	228 [165–347]	185 [161–250]	0.34
**ATIII. % [IQR]**	100 [94–120]	90 [75–104]	0.02
**D-dim. ng/mL [IQR]**	2426 [646–7500]	2677 [908–8000]	0.61
**CK-R time. Min [IQR]**	6.6 [5.2–7.3]	6.8 [5–8.3]	0.19
**CK-K time. Min [IQR]**	1.3 [1.1–1.5]	1.1 [0.8–1.7]	0.20
**CK-Ang. Degrees [IQR]**	73.5 [71.5–75]	75.2 [69.5–78.1]	0.13
**CK-MA. mm [IQR]**	66.1 [64.4–68]	69.6 [68.5–70.7]	** *0.006* **
**LY30. % [IQR]**	0	0	-
**Rt-R. Min [IQR]**	0.3 [0.2–0.4]	0.3 [0.2–0.4]	0.33
**Rt-K. Min [IQR]**	0.8 [0.7–0.9]	0.7 [0.6–0.8]	** *0.002* **
**Rt-Ang. Degrees [IQR]**	79 [78.3–81.6]	81 [80–82]	** *0.001* **
**Rt-MA. mm [IQR]**	67.1 [64.6–68.9]	71 [68.9–71.8]	** *0.001* **
**TEG-ACT time. Sec [IQR]**	87.9 [69.2–97.3]	78.5 [69.2–87.9]	0.32
**CRT-A10. mm [IQR]**	64 [60–67.3]	70 [67.4–71]	** *0.001* **
**CKH-R time. Min [IQR]**	6.3 [4.5–6.6]	6.3 [5.5–7.5]	0.50
**CKH-K time. Min [IQR]**	1.1 [0.9–1.3]	1.1 [0.8–1.3]	0.55
**CKH-Ang. Degrees [IQR]**	76 [73.4–77.1]	76 [75–79]	0.33
**CKH-MA. mm [IQR]**	66 [64.6–68.4]	69.9 [67.7–70.4]	** *0.001* **
**FF-MA. mm [IQR]**	31 [28–39]	46.3 [40–53]	** *0.0004* **
**CFF-A10 mm [IQR]**	30 [26–37]	41 [36–45.5]	** *0.002* **

Abbreviations: *LUS, lung ultrasound*; DIC, disseminated intravascular coagulation; SOFA, sequential organ failure assessment; aPTT, activated partial thromboplastin time; INR, international normalized ratio; FIB, fibrinogen; ATIII, antithrombin III; D-dim, D-dimer; CK, citrated recalcified kaolin-activated blood; RT, rapid thromboelastograph; ACT, activated clotting time; CRT, citrated recalcified kaolin and tissue factor activated blood; A10, amplitude 10 min after clotting time; CKH, citrated recalcified kaolin-activated blood treated with heparinase; FF, functional fibrinogen; CFF, citrated functional fibrinogen.

## Data Availability

Not applicable.

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
