# Peer review of "Thromboelastography Profile Is Associated with Lung Aeration Assessed by Point-of-Care Ultrasound in COVID-19 Critically Ill Patients: An Observational Retrospective Study"

_healthcare, 2022, doi:10.3390/healthcare10071168_

Round 1

Reviewer 1 Report

Dear authors, thank you for conducting the study and writing the paper. 

Indeed, it's an interesting manuscript. 

However, I have some comments:

you should describe in your methods the laboratory values you collected and the time for the collection. 

you could perform a power analysis to strengthen your results. 

you should avoid repletion of the TEG parameters (see pdf-file).

the rapid-TEG does according to my knowledge not reflect the intrinsic, but the extrinsic pathway (tissue factor) please check and correct. 

the conclusion does not come directly from your results, please revise.

Thank you

Author Response

Dear Esteemed Editors and Reviewers,

Thank you very much for considering our paper for publication in Healthcare Journal and for the opportunity to revise it, making it as much effective as possible.

Here below you can find a point-by-point response to reviewer #1.

All changes in the manuscript were made in red.

Rewiever 1

  • Describe in your methods the laboratory values you collected and the time for the collection

Thanks a lot to the reviewer for this comment. We added the requested items to the methods section. “…Demographics and relevant comorbidities were recorded within the first 24 hours of ICU admission. Blood samples for standard hematological parameters, including platelet count, were obtained from an arterial line on ICU admission. Two blood citrate samples (3 ml) were also collected for laboratory standard coagulation tests such as Activated Partial Thromboplastin Time (aPTT), prothrombin time (PT), International Normalized Ratio (INR), D-dimer levels, and Antithrombin. DIC score per International Society on Thrombosis and Haemostasis (ISTH) criteria and Sequential Organ Failure Assessment (SOFA) score, were calculated for each patient on ICU admission. Heart rate, level of consciousness, oxygenation, and respiratory rate were recorded on admission. The ratio of SpO2 to the fraction of inspired oxygen (SpO2/Fio2), arterial blood gases (PaO2; PaCO2), pH, PaO2/FiO2 ratio were also recorded. A chest X-ray was also taken on admission and its findings were also collected…”

  • Perform a power analysis to strengthen your results. 

This is study is clearly underpowered. Even if a post-hoc power analysis is not completely correct since it may be affected by several biases, we performed the a posteriori power analysis using Fisher’s exact test comparing two independent proportions as per reviewer’s request. As expected, power analysis showed that this study is clearly underpowered. However, to our knowledge, as already specified in the manuscript, no previous studies have been published to assess relationships between ultrasound-based lung aeration and coagulation abnormalities on point-of-care testing in critically ill COVID-19 patients. Therefore, this is just a pilot study whose findings should be interpreted as purely exploratory and descriptive.

For this reason we have further clarified this point in the ‘limitation’ section as follows: “…These observations on a small cohort of 40 patients need confirmation by larger trials aimed at comparing LUS score and FF-MA values with lung tissue density and thromboembolic events determined through chest computed tomography. The findings of this pilot study should be interpreted with caution, considering its purely exploratory and descriptive nature…”

  • Avoid repletion of the TEG parameters (see pdf-file)

Thanks a lot. We have modified the description of table 1 as suggested in the pdf-file.

  • The rapid-TEG does according to my knowledge not reflect the intrinsic, but the extrinsic pathway (tissue factor) please check and correct.

Thank you very much for this comment. It was probably a typing error.

We have modified Line 112 accordingly as follows: “... Rapid TEG (CRT), which describes the extrinsic pathway using both kaolin and tissue factor, allowing a quicker assessment of clot strength, without assessment of clot initiation…”.

  • The conclusion does not come directly from your results, please revise

Thank you very much for this comment. We have modified conclusion section by adding this period: “…Our data suggest that in COVID-19 respiratory failure a loss of lung aeration assessed using LUS may be associated with a hypercoagulability state as defined by the high FF-MA values…”

  • Line 28: platelet function s not given by TEG

Thank you very much. We have modified that sentence as follows: “The hypercoagulable profile as defined by elevated FF-MA and CK-MA may be associated with a low degree of lung aeration as assessed by LUS

  • Line 61: D-Dimer. Corrected with d-dimer
  • Line 62: radiological lungs involvement as modified in radiological disease extension as requested

  • Line 82: below you stated that informed consent was waived??? Please make it clear. […] established diagnosis of heart failure, interstitial lung disease of further etiology, home long-term oxygen therapy, denied informed consent.[…]

Thank you. There was a typographical error on line 268. The consent was obtained according to local IRB recommendations.

  • Line 106: when were the samples taken and how frequently? Was it one per patient, only? What other samples were taken? Please be precise…

Thank you. We have clarified this point as follows “We analyzed the first sample collected per patient, within 24 hours after ICU admission...”

  • Line 130: why not? This could have been done, looking at X.ray data…

We do not agree. To our knowledge no data have been published focusing on our primary endpoint so we were not able to calculate a priori sample size.

  • Line 133: We have modified the text as per your suggestions as follows: “LUS score > 12 was considered as associated with worse severity of the pulmonary disease
  • Line 152: “…maximum amount of clot strength…” modified in “…maximum clot strength…”

  • Line 165: n?? please add the patient numbers. We do not understand

  • Line 169: please remove this here. Either refer to a reference article for explanation or give it in the method section.
    Please avoid this repetition, either in the methods or here…

Done. It was modified as per your suggestion and described above.

  • Fig 1: you did not mention other lab-values had been taken… We do not understand. Figure 1 does not refer to any lab value.

  • Line 199: where is the data? The results given in this sentence reflect fibrinogen level/function but not platelet function…

Thank you. We have modified as follows: “…mainly related to fibrinogen levels and function, as evidenced by elevated FF-MA values…”

  • Line 242: this conclusion does not come from your results. You may put this to the discussion, but these was no proof for such a statement.

Already modified as reported above.

Reviewer 2 Report

The article entitled “Thromboelastography Profile is Associated with Lung Aeration Assessed by Point-of-Care Ultrasound in COVID-19 critically Ill Patients: An observational retrospective study” is a observational retrospective study that analyzes the association between aeration lung by Lung Ultrasound (LUS) and compliance of the respiratory system (Crs) and between LUS and coagulation activation by thromboelastography (TEG) in 40 critically ill patients with COVID-19. The authors found an significant correlation between LUS score > 12 LUS and Crs ≤ 37 and between LUS score > 12 and 46.3 mm of functional fibrinogen maximum amplitude (FF-MA) and kaolin-activated maximum amplitude (CK-MA), respectively. In summary, this study shows that LUS and coagulation profile can be used to diagnosis the severity and prognosis of COVID-19. The purpose of this article is well addressed and the information may be useful in clinical practice. Therefore, I think that this article is suitable for publication in its current version.

Author Response

Thank you very much for your comment

Reviewer 3 Report

Biasucci et al submitted a manuscript entitled “ TEG profile is associated with LUS by point of care ultrasound in COVID-19 critically ill patients: an observational retrospective study”.

In general, the subject is interesting and the approach both timely and original.

I first wonder if Healthcare journal ‘s field is adapted for the purpose of this study. Editorial team will judge about this.

However, many limitations can be reported as listed:

As major limitations:

  • I really wonder about the design of the study described as a "retrospective analysis of prospectively and systematically collected data”. First because authors mentioned “denied informed consent” (prospectively collected ?); second because ultrasound data collection and LUS score cannot be collected retrospectively easily and in a routine care and finally, I wonder if TEG6 real time analysis is a routine test in their hospital (involving a direct analysis) ?
  • Line 51 : authors mentioned chest radiography in comparison to US. However, I cannot agree with the proposition that US (compared to chest radiography) has the advantage of easy repeatability (high operator dependence for US), short duration (shorter for chest radiography) and minor operator ‘exposure to the virus (the operator is directly in contact with the patient !) : please review this sentence.
  • According with low effective and considering that all values (exception for a few of them) are presented with IQR, and to simplify, I highly recommend presenting all values with median IQR and to only perform non parametric tests
  • Antithrombotic use was a well-adapted exclusion criteria. However, authors do not mention anticoagulant and their position. Critically ill patients are patients with a high frequency of anticoagulant use, even in a prophylactic indication. Please be accurate on this
  • As the first limitation: I am not sure to appreciate the interest of thrombotic risk by LUS (exposing operator to virus with non-reproducible procedure) if the TEG is a biological (standardized procedure with a real-time results). Please discuss it and adapt both discussion and conclusion (line 244-245).

As minor limitations:

  • Crs line 22 need to be defined
  • Line 99 to 105 : could the authors provide image illustrating a LUS score of 0;1; 2 and 3 to meet the reader
  • All table 1 legend : line 169 to 184 should be placed to method section
  • Table 1 : aPTT is not define : please check it

Thanks to the authors and the editorial team for this review request.

Author Response

  • Design of the study: retrospective analysis of prospectively and systematically collected data”.

- Author mentioned “denied informed consent” (prospectively collected?);

- Ultrasound data collection and LUS score cannot be collected retrospectively easily and in a routine care

- I wonder if TEG6 real time analysis is a routine test in their hospital (involving a direct analysis)?

In our institution, lung ultrasound is routinely performed in acute respiratory failure, on a daily basis. TEG6s is performed in septic patients and we started to perform in COVID-19 patients during the first wave. We retrospectively reviewed all records in which both LUS and TEG6s were performed within the first 24 hours after ICU admission.

We have modified the text accordingly.

  • Line 51 : authors mentioned chest radiography in comparison to US. However, I cannot agree with the proposition that US (compared to chest radiography) has the advantage of easy repeatability (high operator dependence for US), short duration (shorter for chest radiography) and minor operator ‘exposure to the virus (the operator is directly in contact with the patient !) : please review this sentence.

We have modified the sentence as follows: “…used as an alternative to chest radiography for diagnosis and monitoring of COVID-19 related interstitial pneumonia and acute respiratory distress syndrome, with the advantages of easier bedside use, easy repeatability, immediate availability, fewer operators  exposed to the virus and to ionizing radiations during the single exam…”

  • According with low effective and considering that all values (exception for a few of them) are presented with IQR, and to simplify, I highly recommend presenting all values with median IQR and to only perform nonparametric tests.

We do not agree.

  • Antithrombotic use was a well-adapted exclusion criteria. However, authors do not mention anticoagulant and their position. Critically ill patients are patients with a high frequency of anticoagulant use, even in a prophylactic indication. Please be accurate on this.

This was beyond the scope of the present study.

  • As the first limitation: I am not sure to appreciate the interest of thrombotic risk by LUS (exposing operator to virus with non-reproducible procedure) if the TEG is a biological (standardized procedure with a real-time results). Please discuss it and adapt both discussion and conclusion (line 244-245).

Also this was not within the scope of our study. The final message is these two point-of-care diagnostic modalities may be useful for bedside severity assessment and monitoring of critically ill COVID-19 patients as already pointed-out in the conclusion section.

Minor.

  • Crs line 22 need to be defined

Thank you very much. It has been defined.

  • Line 99 to 105 : could the authors provide image illustrating a LUS score of 0;1; 2 and 3 to meet the reader.

We have not enough space for this. We have quoted two references at least in which the LUS score adopted has been extensively described.

  • All table 1 legend : line 169 to 184 should be placed to method section.

Done it as per your suggestion. Thank you very much.

  • Table 1 : aPTT is not define : please check it

Thank you very much also for this. We have clarified this abbreviation in the text.

Reviewer 4 Report

From my point of view, the manuscript is concise, but – as written by the authors - to the present knowledge, no data have been published yet describing the relationships between LUS findings and procoagulant profiles in COVID-19 adult critically ill patients.

Thus, I consider the data original and highly appreciate these results that can easily and quickly help to the decision of the physician taking care for the patient. This way, they contribute to the knowledge about this disease improving the care for the COVID-19-affected patients.

Taken everything into account, according to my opinion, the article can be published in the current form.

Author Response

Thank you very much

Round 2

Reviewer 3 Report

As a second submission, authors well improved the quality of the manuscript accordingly with my previous remarks. 

The MS can now be considered for publication

Thanks to editorial team and authors for this review request